# Uganda Mountain Community Health System—Perspectives and Capacities towards Emerging Infectious Disease Surveillance

**DOI:** 10.3390/ijerph18168562

**Published:** 2021-08-13

**Authors:** Aggrey Siya, Richardson Mafigiri, Richard Migisha, Rebekah C. Kading

**Affiliations:** 1Department of Environmental Management, Makerere University, Kampala P.O. Box 7062, Uganda; 2EcoHealth180, Kween District, Kapchorwa P.O. Box 250, Uganda; 3Global Health Department, Infectious Diseases Institute, Makerere University, Kampala P.O. Box 22418, Uganda; rmafigiri@gmail.com; 4Department of Physiology, Mbarara University of Science and Technology, Mbarara P.O. Box 1410, Uganda; rmigisha@must.ac.ug; 5Department of Microbiology, Immunology, and Pathology, Colorado State University, Fort Collins, CO 80523, USA; rebekah.kading@colostate.edu

**Keywords:** alerts, village health teams, community-based surveillance, integrated disease surveillance and reporting, Elgon, climate change, one health

## Abstract

In mountain communities like Sebei, Uganda, which are highly vulnerable to emerging and re-emerging infectious diseases, community-based surveillance plays an important role in the monitoring of public health hazards. In this survey, we explored capacities of village health teams (VHTs) in Sebei communities of Mount Elgon in undertaking surveillance tasks for emerging and re-emerging infectious diseases in the context of a changing climate. We used participatory epidemiology techniques to elucidate VHTs’ perceptions on climate change and public health and assessed their capacities to conduct surveillance for emerging and re-emerging infectious diseases. Overall, VHTs perceived climate change to be occurring with wider impacts on public health. However, they had inadequate capacities in collecting surveillance data. The VHTs lacked transport to navigate through their communities and had insufficient capacities in using mobile phones for sending alerts. They did not engage in reporting other hazards related to the environment, wildlife, and domestic livestock that would accelerate infectious disease outbreaks. Records were not maintained for disease surveillance activities and the abilities of VHTs to analyze data were also limited. However, VHTs had access to platforms that could enable them to disseminate public health information. The VHTs thus need to be retooled to conduct their work effectively and efficiently through equipping them with adequate logistics and knowledge on collecting, storing, analyzing, and relaying data, which will improve infectious disease response and mitigation efforts.

## 1. Introduction

Emerging and re-emerging infectious diseases continue to threaten human health across the globe. The current COVID-19 (coronavirus disease 2019) pandemic exemplifies the wider impacts of such diseases on different aspects of the economy, human livelihoods, and well-being [1,2,3]. It is thus crucial that communities are prepared for the next pandemic to minimize such impacts. Estimates indicate that about three quarters of the emerging and re-emerging infectious diseases are zoonotic in nature [4,5]. A zoonotic disease is an infectious disease that jumps from a non-human animal to humans through direct contact or through food, environment, water, or unconventional agents [4]. These diseases present a major public health crisis because of the intricate relationship between humans and animals through, for example, food production and distribution practices, as well as the interconnectedness of human populations [6].

Whereas domestic and wild animals are important hosts for zoonotic disease pathogens, human encroachment on the ecological systems has exacerbated spillover of such pathogens to humans [7]. Pathogen spillover from an animal reservoir to a human is a complex process involving many barriers at the reservoir, environment, and host levels; when these barriers are weakened in such a way that increases pathogen pressure in the reservoir or environment, exposure at the animal–human interface, or susceptibility to infection, the probability of spillover becomes even more likely [8]. Common drivers of pathogen emergence include economic development and land use changes, international travel and commerce, human behaviour and demographics, changes in ecosystems, changes in human susceptibility, and hospital-associated events [8]. Disruption of the ecosystems by humans has been occurring in the past decades with an increase in the recent past [9,10]. For instance, between 2004 and 2014, it was estimated that Ebola disease outbreaks in Central and West Africa correlated with hotspots of forest fragmentation [11]. Similarly, highly fragmented landscapes in Bangladesh were found to be high-risk areas for Nipah virus disease outbreaks [12]. The recent ecosystem disruptions have mainly been fueled by efforts to ensure a sustained food supply through agricultural production, processing, and distribution [9,13], but consequently have also increased contact at the human–animal interface. The human population is expected to reach 9.7 billion by 2050; 2.5 billion of this population will be in Africa [14], and urbanization characterized by the rapid intensification of agriculture, socioeconomic change, and ecological fragmentation will occur [15]. Projections on food demands indicate a 60% increase in sub-Saharan Africa [16,17] implying increased ecosystem disruptions and an associated increase in vulnerability at the human–animal interface that may weaken barriers to spillover and facilitate the movement of disease-causing pathogens to humans.

Human influence has also accelerated climate change through land use changes and increased greenhouse gas emissions, further disrupting wildlife ecosystems and increasing the human–animal interface [18,19]. The changing climate will have both direct and indirect consequences not only on the spillover events of such pathogens, but also on their transmission within and among human populations. For example, climate change may accelerate range shifts of different species that could be natural hosts of pathogens that can spillover to humans, causing disease [20]. Besides effects on the distribution of natural hosts, climate change might also undermine food security, presenting additional challenges for disease management in a health infrastructure that is already stressed or underdeveloped. These pressures have recently been witnessed during the management of SARS-CoV-2 (Severe acute respiratory syndrome coronavirus) response efforts, especially in Africa where most households live on “a hand to mouth” strategy where one has to look for what to eat every day and nothing gets stored for future consumption [21,22,23]. In mountain communities that have been cited to be among the highly sensitive groups to climate change, such infectious disease scenarios will be pose fundamental impacts on human populations [24]. This impact has been evidenced for other diseases like malaria whose belt has been documented to expand in mountain areas like Mount Elgon [25,26,27]. Climate change tends to alter the environmental conditions providing suitable habitats for disease vectors, e.g., malaria. Rapid urbanization that is occurring in these mountain areas will further accelerate the disruption of ecosystems, increasing both the human–animal interface and disease pathogen spillover events [28]. Furthermore, climate change together with other factors like land degradation on mountain slopes may result into heavy rains causing disasters like the landslides evidenced recently evidenced in Mount Elgon [29]. Such disasters can facilitate disease outbreaks associated with water, e.g., cholera. Collectively, these pressures call for the design of public health interventions that are sensitive to mountain communities during the management of infectious disease outbreaks.

Uganda has received credit for supporting the management of disease outbreaks in other countries including the 2014–2015 Ebola virus disease outbreak in West Africa [30]. The country’s efforts to prevent spillover across national borders of the recent Ebola outbreak in the neighboring Democratic Republic of Congo strengthened such recognition. Despite this credit, Uganda continues to experience regular disease outbreaks of diverse pathogens including yellow fever [31], tuberculosis [32], measles [33,34], and hepatitis E [35], as well as the re-emergence of diseases after many years, like Rift Valley fever [36]. Moreover, some of these disease outbreaks are always recurring and have a pandemic potential. Based on the previous experiences, it is expected that the country’s health system has greatly improved to prevent spillover events, ensure early detection, and facilitate an effective response to disease such diseases. Unfortunately, there seems to be inadequate capacities of the health system to prevent, detect, and respond to such disease events. Previous disease outbreaks were identified through detecting an abnormal increase in deaths and severe signs and symptoms presented by humans. This approach is not adequate as it does not increase potential for detection of an index case. The early detection of index cases is critical to deploy early actions to prevent the escalation of the disease and its spread to different geographic areas.

In mountain communities that often have inadequate infrastructure for relaying information, community-based disease surveillance requires adequate empowerment to navigate through hard-to-reach areas. Despite the occurrence of eight infectious disease outbreaks in Uganda in 2017–2018 alone, viral hemorrhagic fevers were not reported according to the International Health Regulations, which creates a significant public health and biosecurity threat [37]. Besides infrastructure, the changing climate and human demographic characteristics increases their vulnerability to infectious disease outbreaks [24]. In countries like Uganda, mountain communities like Sebei in Mount Elgon have witnessed disease outbreaks that were never reported before. These, among others, include the following: anthrax [38], polio [39], cholera, and Marburg virus [40]. Besides, the malaria belt has expanded in the area [27], thus exemplifying the need for enhanced interventions to prevent further disease outbreak events. VHTs report to the nearest health facility that serves the catchment area where they reside, and they are sometimes referred to as community health volunteers. Their roles, among others, include the mobilization of communities for improved health service access. They also contribute to the management of diseases. They thus provide an important avenue for understanding surveillance capacities of the communities that they represent.

This survey thus sought to elucidate capacities of the village health teams (VHTs) in ensuring community-based surveillance in the Kween District areas of Mount Elgon.

## 2. Materials and Methods

### 2.1. Study Site, Study Population, and Sampling Strategy

This study was conducted as part of the Mountain Sentinels fellowship program that involved enhancing the resilience of mountain communities during the COVID-19 pandemic, which has widely undermined the livelihoods of communities. During this fellowship, a research project on enhancing community-based surveillance for infectious diseases in the Sebei community of Mount Elgon was proposed (Figure 1). This was an action-oriented research project that involved training of communities on emerging and re-emerging infectious diseases and their causes, including climate change and environmental degradation, as well as providing information about community-based disease surveillance and its importance. The engagement targeted all the communities living in all the altitudinal zones of the mountain. Kween has recently experienced several infectious disease outbreaks including Marburg, anthrax, cholera, foot and mouth disease, and measles [38,40]. The changing climate has led to the expansion of the malaria belt towards the higher altitudes. Besides infectious diseases outbreaks, the area experiences landslides and the lasting effects of previous conflicts from the neighboring communities of Karamoja and Pokot. Disease surveillance in such areas thus ought to integrate such aspects as per the World Health Organization (WHO, Geneva, Switzerland) guidelines.

This study was conducted in 10 purposively selected villages in the Kween District in the Mount Elgon areas of Uganda during the months of November and December 2020. The villages were targeted to cover different altitudinal zones of the mountain with different socioeconomic activities. Notably, the villages were selected in areas near the Mount Elgon national park, a mid-altitudinal zone of the mountain with heavy human settlement, and finally, in the lower belts with vast grasslands and flat land. Additionally, the selection of villages was based on the availability of a VHT member. Those without VHT members were excluded from the study. The village health teams (VHTs) that form part of the community health care workers (at the “Health Center I” according to the structure of Uganda’s health system) of the Ugandan health system were selected using convenience sampling techniques based on their availability for the engagement. The VHTs were selected because they stay and work closely with their communities and have the mandate to report health-related events within their communities.

### 2.2. Data Collection

A semi-structured interview questionnaire as used for the collection of qualitative and quantitative data. The questionnaire consisted of three sections: (1) socio-demographic characteristics, (2) perceptions on climate change and public health aspects, and (3) infectious disease surveillance capacities (including collection, analysis, and dissemination). The questionnaires were administered to the focus group discussion members that formed part of the team engaged. There were 10 focus group discussions held in the study area, with members in each group not exceeding five and not less than three. Written informed consent was sought from them about their willingness to participate in the interview. This was done after explaining to them about the study and the activity broadly.

### 2.3. Scoring of Perceptions

Perceptions on climate change and public health as well as infectious disease surveillance capacities that were answered positively were scored one, while those that were answered negatively were scored zero. All questions were given equal weight, and missing responses were not scored. Meanwhile, responses where individuals had no knowledge about a particular aspect were scored two. The scores that were made during the recording of responses were not directly used in the computation of the percentage scores of each response. The scores for the different responses were coded again and percentages were computed to obtain the actual score.

### 2.4. Data Analysis

Data were analyzed using previously described methods, with some modifications made [41]. Data captured as a score, such as responses regarding different aspects of climate change and surveillance, are reported as means and percentages (Figure 2, Figure 3 and Figure 4). Meanwhile, qualitative data were transcribed and analyzed using thematic analysis to understand people’s opinions regarding climate change and public health as well as disease surveillance. Themes on disease surveillance components of data collection, analysis, and dissemination were explored among the village health teams to elucidate their capacities in undertaking disease surveillance tasks. The themes on climate change and its public health impacts were also explored. Thematic analysis was conducted using NVivo Version 12 (https://www.qsrinternational.com/nvivo-qualitative-data-analysis-software/, access on 23 March 2021), a software program (QSR International, Melbourne, Australia) that is often used for acquiring deep insights regarding qualitative data that can be in the form of unstructured text.

## 3. Results

### 3.1. Sociodemographic Characteristics

In total, 48 village health team professionals were interviewed from 10 different villages throughout the Kween District. The 10 villages represented the main villages just before their subdivision, and the 48 VHTs were derived from the subdivisions of the main villages. Some villages had more than one VHT member and they were enrolled into the study. Focus groups from each village comprised 3–5 participants from the subdivisions of the villages. The participants reared livestock (mainly goats and cattle) and had access to nearby schools (Table 1). All utilized both modern medicine and services from traditional healers.

### 3.2. Perceptions on Climate Change and Impacts on Public Health

#### 3.2.1. Rainfall and Temperature

Respondents from each of the 10 villages were in complete agreement that rainfall onset and cessation of rainy seasons are highly variable. They also noted that quantities of rain were abnormally high/low during some periods with more flooding experienced. Dry periods during the rainy season were noted to be more common and lengthy now compared to the past. Respondents from 9 out of 10 villages agreed that droughts came more often and that hailstorms were more frequent.

Similarly, all the respondents agreed that temperature in both rainy and wet seasons has increased.
*“These days, the rainfall has greatly changed. Sometimes we receive abnormally large quantities of rainfall and sometimes it gets extremely low. The onset and cessation of these rains have greatly changed and hard to predict”*. Noted by a VHT member.
*“Temperature sometimes gets so high that even our water wells dry up. This is common now unlike in the past”.* Noted by a VHT member.

#### 3.2.2. Climate Change Events

Frost was noted to be more common and intense now affecting crop produce especially in areas near the Mt Elgon National park. The fog, especially in areas near the park, was noted to have reduced. Flooding within the region was noted to have increased in the area and was attributed to the change in climate by survey participants
*“These days, floods are more common and especially in the flat areas of Greek, Ngenge and Sundet. This was not the case in the past because these rains at times come when they are too much”.* Noted by a VHT member.

#### 3.2.3. Climate Change Impacts on Health

Regarding the impacts of climate change on health, respondents generally agreed that climate change has led to the expansion of the range of vectors that transmit diseases causing pathogens. An example that was raised was the mosquitoes that are now common in some higher altitude areas, causing malaria. The new diseases that affect livestock and humans were also noted to be common now because of the change in climate conditions. The respondents went ahead to list examples of diseases that are emerging within their community because of climate change including Marburg virus disease, anthrax, and foot and mouth disease. Food systems were also noted to have been disrupted by changes in the climate.
*“The malaria cases are now common in cold areas near Mt. Elgon national park and yet they were not there in the past. The mosquitoes are moving to these areas and it is because of the warming temperature there”.* Noted by a VHT member.
*“New diseases are now common that affect humans and livestock. These diseases include foot and mouth disease, Marburg, anthrax, and cholera. These diseases are coming up because of the changing climate associated with floods and high temperature”.* Noted by a VHT member.
*“We experience many environmental events like mud sliding, flooding of the rivers and yet this would cause diseases and also destroy our crops”.* Noted by a VHT member.

### 3.3. Surveillance Capacities

#### 3.3.1. Data Collection

Regarding data collection capacities, all the VHTs had emergency contacts for reporting public health events within their communities to the government authorities (Figure 2). Up to half of the VHT participants owned private mobile phones and they had been trained on the signs and symptoms of common infectious diseases within their communities. Only 40% of the respondents knew how to use mobile phones to send text messages or even use them for voice calls. Similarly, only 40% of the respondents had access to the nearby health facility, as only 10% of total number of respondents had a means of transport (Figure 2). Regarding record keeping, only 20% of the respondents noted to keep records of their activities. Regarding the kind of diseases reported that may not occur in humans, only 30% of the VHTs reported diseases that affect their domestic livestock, while none of them reported diseases in wildlife and environmental hazards like floods (Figure 2).
*“I have a mobile phone but I am not well-versed with typing messages. I sometimes just give a friend/my child to type and send the message. But it is too much work because I have to explain to the person helping me to type and yet there is no payment for the service”.* Noted by a VHT member.

#### 3.3.2. Data Analysis

All VHTs that engaged knew how to identify the signs and symptoms of common infectious diseases within their community (Figure 3). Meanwhile only 10% of them knew how to analyze the data they collected using bar graphs, while none of them knew how to draw line graphs for the disease trends within their communities.
*“We have been taught several times about the signs and symptoms of common diseases that’s why we are able to identify such patients in the community”* Noted by a VHT member.

#### 3.3.3. Information Dissemination

Regarding dissemination of information, all VHTs that engaged had access to a venue for dissemination of health-related information (Figure 4). All VHTs also had stationery for training and dissemination of public health information and were part of the social group that can allow for the dissemination of public health information. Access to radio stations and nearby schools for the dissemination of public health information was limited for up to half (50%) of the respondents (Figure 4).
*“I do community walks sometimes and meet with my community members. I can pass on the message on public health events while paying them a visit”.* Noted by a VHT member. *“I am part of the village savings team and sometimes I disseminate health-related messages to my teammates whenever we meet. I would reach out to the broader community but it is hard to reach out to all areas as it needs a transport means”*. Noted by a VHT member.

## 4. Discussion

Here, we present the first study that focuses on community-based disease surveillance in the changing climate within the mountain communities of Uganda. Specifically, we employed qualitative approaches to assess public health perceptions and disease-reporting infrastructures among rural mountain communities in Eastern Uganda that are critical for responding to emerging infectious disease events. This area has experienced increasing infectious disease outbreaks in recent years, and increased incidence of highland malaria consistent with the influence of climate change on mosquito vector distribution [25,26,27,38,40,42]. The ability to prevent, detect, and respond early to these events at the village level is paramount. To this end, we interviewed village health team workers from 10 villages in the Kween District to gain a preliminary understanding of the peoples’ perceptions on climate change and infectious diseases, and importantly, to determine what shortcomings are present in the disease-reporting infrastructure that could be improved for more effective public health protection.

Previous studies on community perceptions regarding climate change within the Mount Elgon areas of Kapchorwa also occupied by the Sebei community indicated a highly variable rainfall onset and offset recently [43,44,45]. This result is similar to that of this current study, in which VHTs (who are part of the community) perceived fluctuations in important climate variables (rainfall and temperature to be highly variable currently). Additionally, climate events including drought and highly erratic rainfall resulting in floods and reduced food access were also noted in this current study. Although most of these studies related climate change with food security, climate change is generally perceived to be occurring. Previous data within the Kween District and in the whole of Mount Elgon indicated highly variable rainfall and temperature recently compared to the past [27]. This is consistent with opinions from the VHTs in this current study regarding climate trends. However, the less significant result regarding climate change aspects of rainfall in Mount Elgon in other studies could be because the methods used do not take into account human activities while making predictions [46]. Existing studies on climate change perceptions in Mount Elgon, however, did not focus much on diseases and the surveillance/health system. The neighboring pastoral communities of Karamoja and Teso have similar perceptions of climate change with a highly variable rainfall onset/offset and longer and frequent hot periods [47,48]. Meanwhile, elsewhere within East Africa, a study on perceptions regarding climate change in Zanzibar indicated climate change to be occurring with increased rainfall and extreme temperature affecting sea levels and livelihoods [49]. Furthermore, other studies have indicated similar results [50,51]. Climate adaptation plans ought to be designed taking into consideration location perceptions. Notably, local perceptions and interventions should be integrated in the wider climate plans so as to realize optimal impact. In doing so, actions can easily be taken by the community as they already know what is going on and understand firsthand the consequences if they do not change their actions. Negotiating such actions can be driven through in a food security lens, as communities appear to understand more about climate change from a food security perspective.

Whereas VHTs have been shown to play a fundamental role in health care delivery, they still grapple with challenges. In this study, we demonstrated that VHTs have challenges regarding data collection, analysis, and dissemination of public health information. Specifically, the VHTs during data collection cannot navigate their localities due to poor infrastructure and inadequate means of transport. They also do not have adequate capacities to store data and relay alerts using their mobile phones. This result is similar to the findings of a study conducted in the semi-arid areas of Pader in Northern Uganda, where VHTs complained about inadequate transport to reach out to their communities [52]. Other studies have indicated logistical issues in the form of transport and communication to be the main challenges hindering VHTs from undertaking their tasks [53,54]. Other studies have also indicated inadequate trainings to be another challenge faced by VHTs, which is similar to what was found in this study [55]. This study revealed the inadequate training of VHTs on the signs and symptoms of common diseases in their locality. The low levels of knowledge regarding data analysis also indicated inadequate trainings regarding such aspects. However, the VHTs may not even have the capacities to analyze data because of the challenges with recruitment. Some VHTs are recruited by their peers even when they do not have the capacities to undertake disease surveillance tasks. Studies have indicated that recruitment of VHTs is not formal and often undermines their performance [56,57,58,59]. The lack of participation of VHTs in reporting of wildlife/animal-related disease and environmental events, as indicated this study, provides an opportunity for diseases to escalate and spillover to humans from livestock and wildlife. There is need to integrate such aspects into the scope of work for the VHTs to enhance pandemic preparedness. Other review studies have recommended similar aspects so as attain optimal health of not only humans, but also animals, thus minimizing the chances of future pandemics [60].

The Integrated Disease Surveillance and Response (IDSR) strategy was adopted by Uganda (from 2010) to adequately manage disease outbreaks and requires the integration of the community health and One Health aspects to achieve the intended goals. This strategy will increase contributions to the achievement of the goals of International Health Regulations (IHR, 2005), the Global Health Security Agenda (GHSA), and One Health. The implementation of a One Health framework for infectious disease response in Uganda has been met with mixed success, having achieved national-level strategic planning and recognition, but facing challenges, such as the need for strengthened intersectoral coordination on the ground [61]. The IDSR/IHR committee and the National Task Force on Epidemics (NTF) at the national level, and the Emergency Preparedness and Response Committee (EPRC) and the District Task Force (DTF) at the district level, constitute the structures responsible for IDSR implementation in Uganda, which ought to have VHTs on board to facilitate the relaying of their concerns and redresses, as well as to execute strategic health initiatives at the village level. Integration of this community-level surveillance into the national strategy will facilitate monitoring, prevention, and early reporting of disease events.

## 5. Conclusions

Community-based surveillance is very important in preparing for the next pandemic. The village health teams that live closely with the community provide an important avenue for preventing early detection and early response to such public health hazards. They facilitate monitoring of such events, allowing for actions to be taken adequately. However, the VHTs need to be retooled so as to respond adequately to public health hazards. This study has indicated how such key stakeholders in surveillance are not adequately equipped to prevent and manage public health hazards. Moreover, the ones assessed are in a highly vulnerable mountain community that has already experienced emerging and re-emerging infectious diseases. Furthermore, such key stakeholders should be equipped to report on environmental changes that are important risk factors for emerging and re-emerging infectious diseases. Such aspects can include climate-related events and landslides. Relatedly, they need to be champions in emerging environmental conservation so as to minimize environmental degradation. In doing so, the impacts of proposed approaches like IDSR can be realized adequately.

## Figures and Tables

**Figure 1 ijerph-18-08562-f001:**
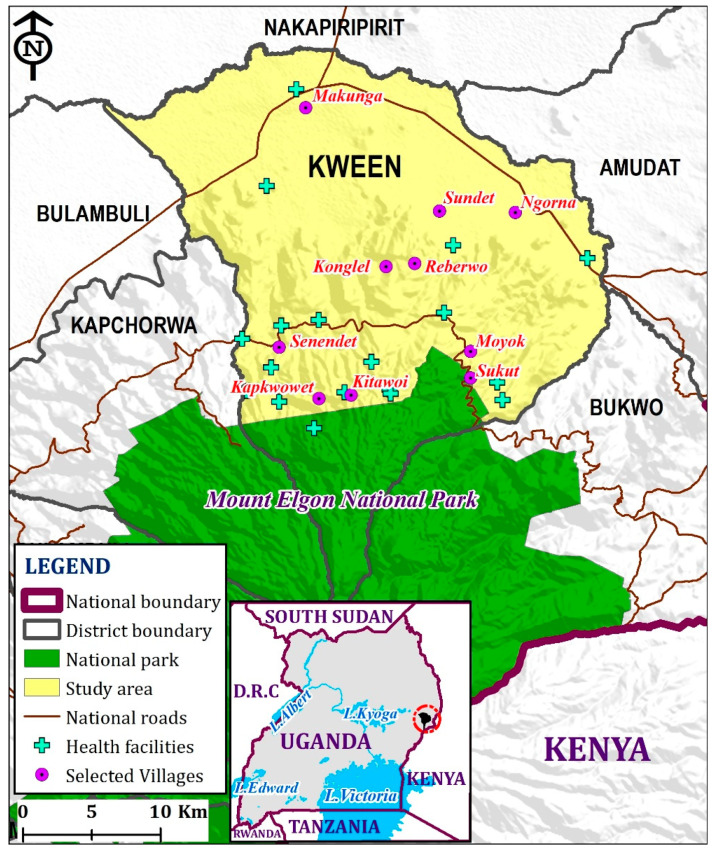
Location of the Kween District.

**Figure 2 ijerph-18-08562-f002:**
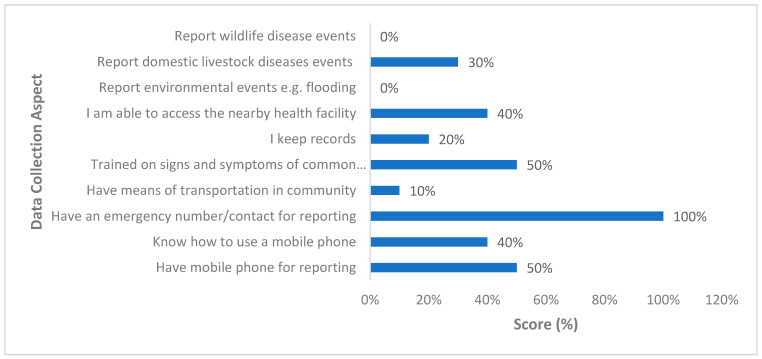
Percentage of village health team members across all 10 villages who responded positively to disease detection and reporting capacities in the Kween District, Uganda, 2020.

**Figure 3 ijerph-18-08562-f003:**
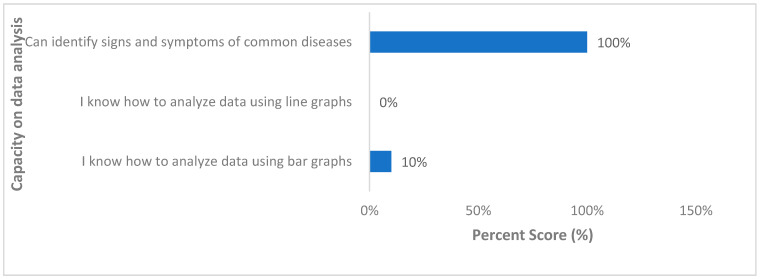
Data analysis capability of village health team members across all 10 villages in the Kween District, Uganda, 2020.

**Figure 4 ijerph-18-08562-f004:**
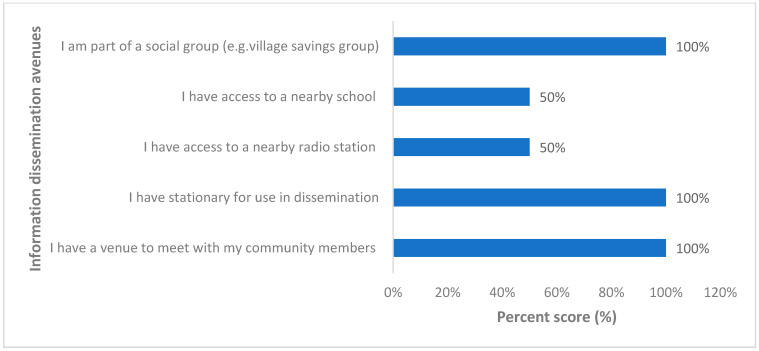
Opportunities for health information dissemination by village health teams in the Kween District, Uganda.

**Table 1 ijerph-18-08562-t001:** Sociodemographic characteristics.

Village *	Konglel	Reberwo	Ngorna	Sundet	Kitawoi	Kapkwowet	Sukut	Senendet	Magunga	Moyok
Average herd size (number of cows)	12 ± 3	14 ± 3	19 ± 2	22 ± 1	6 ± 4	12 ± 9	7 ± 2	12 ± 1	16 ± 3	13 ± 3
Average herd size (number of sheep)	1 ± 1	0	3 ± 5	5 ± 3	0	0	0	0	1 ± 1	2 ± 7
Average herd size goats	16 ± 7	13 ± 1	22 ± 2	26 ± 6	11 ± 3	5 ± 5	14 ± 2	13 ± 4	11 ± 2	12 ± 9
Average farm size (ha)	1 ± 2	1.2 ± 5	12 ± 1	25 ± 3	1 ± 3	1.3 ± 1	1.1 ± 2	1 ± 3	1.2 ± 2	1 ± 2
Level of access to water source point	Medium	Yes	Low	Low	Medium	Medium	Medium	Medium	Low	Medium
Access to electricity	No	No	No	No	No	No	No	No	No	Yes
Number of nearby primary schools	4	3	2	2	4	2	2	1	3	4
Number of nearby secondary schools	2	2	1	1	2	2	2	2	2	2
Where do you access health services? Traditional healer or Western medicine?	Both	Both	Both	Both	Both	Both	Both	Both	Both	Both

* Data are expressed as the mean number ± standard deviation.

## Data Availability

Data for this survey are available upon reasonable request.

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
