# Peer review of "Uganda Mountain Community Health System—Perspectives and Capacities towards Emerging Infectious Disease Surveillance"

_ijerph, 2021, doi:10.3390/ijerph18168562_

Round 1
Reviewer 1 Report
Review of Uganda Mountain Community Health System Perspectives and Capacities Towards Emerging Infectious Disease Surveillance.
This is a well-written paper with important findings which should be communicated widely.
I recommend the manuscript be accepted for publication if the comments made in my attached peer review are addressed.
|
Section |
Page |
Line |
Comment |
|
|
2 |
99 |
Spelling - recognition |
|
|
3 |
123 |
Define VHT |
|
|
3 |
141/147 |
Were the VHTs selected randomly or purposively? |
|
|
4 |
166/7 |
When respondents don’t know about a subject they are questioned on, I disagree with scoring this as a zero (previously state as denoting a negative response). What is your rationale for this? I recommend re-analysing the data taking this into account. |
|
|
5 |
171 |
Briefly describe data analysis method and clarify what the deviations from the previously described method (referenced) are. |
|
|
|
173 |
Table 1.1 is missing |
|
|
|
182 |
Missing ‘the’ from sentence - in the form |
|
Table 3 |
|
|
Mislabelled – should be Table 1 |
|
|
|
|
The row “Where do you access health services?” is merged with the last row and has no data |
|
|
|
192 |
Should read ‘impacts on public health’ |
|
|
|
196 |
Should read rain not rainy |
|
|
5 |
200 |
Correct spelling - respondents |
|
Sections 3.2.2 & 3.2.3. |
|
|
Results need data (% responses) to support them as in other sections. A figure to show key responses should be included as many questions were asked and no data shown. |
|
|
8 |
297 |
Brackets incorrectly placed. |
|
|
|
298 |
Should read in not into |
|
|
9 |
318 |
Delete in |
|
|
|
|
|
Author Response
Thank you very much for the review of our manuscript. We have revised as per track changes and also in the attached table

Reviewer 2 Report
This work addresses a very pertinent and current issue: the tendency for the emergence of new diseases, a consequence of excessive human intervention in nature, focusing on the perception of the inhabitants. The authors study the case of Uganda Mountain Community Health System.
The objectives are correctly defined. The methodology is well-designed and is consistent with the objectives of the study. The interpretation and discussion of results is clear, objective, and consistent, although a deeper analysis of the obtained data could be done. The conclusions summarize well the results obtained and are consistent with the work presented. However, there are some details that can be improved:
- The title would be more noticeable using a colon or dash after “Uganda Mountain Community Health System”;
- Lines 39, 157, 158, 159, 164, 165, 167, 237, 255, 270: It is unnecessary to present the values in numerical format and in full.
- Lines 188 e 189: Just refer to Table 1 once.
- Line 190: Change Table 3.1 to Table 1, as referred to in the text.
Author Response
General Comment:
This work addresses a very pertinent and current issue: the tendency for
the emergence of new diseases, a consequence of excessive human
intervention in nature, focusing on the perception of the inhabitants.
The authors study the case of Uganda Mountain Community Health System.
The objectives are correctly defined. The methodology is well-designed
and is consistent with the objectives of the study. The interpretation
and discussion of results is clear, objective, and consistent, although
a deeper analysis of the obtained data could be done. The conclusions
summarize well the results obtained and are consistent with the work
presented. However, there are some details that can be improved:
Response: We thank the reviewer for this positive feedback and giving us comments to further enhance the quality of our manuscript.
Comment #1
The title would be more noticeable using a colon or dash after “Uganda
Mountain Community Health System”;
Response: We thank the reviewer for this suggestion. We have added an em dash (—) after “Uganda Mountain Community Health System” in the title.
Comment #2
Lines 39, 157, 158, 159, 164, 165, 167, 237, 255, 270: It is unnecessary
to present the values in numerical format and in full.
Response: We thank the reviewer for this observation. We have deleted the numerical figures that were in brackets in the revised version.
Comment #3
Lines 188 e 189: Just refer to Table 1 once.2
Response: We have revised this part of the manuscript and Table 1 is now referenced once.
Comment #4
Line 190: Change Table 3.1 to Table 1, as referred to in the text.
Response: We have revised the heading of the table for socio-demographic characteristics accordingly, to reflect Table 1.
Reviewer 3 Report
In a mountain district of Uganda 48 Village Health Team professionals from 10 different villages have been interviewed by questionnaires whether there is climate change and increased incidences of diseases like malaria. The problem is that the responses are personal opinions of the interviewed which are not verified by data such as temperature measurements, number of patients with the different illnesses as compared to a defined time before. Thus, the manuscript reports individual assumptions rather than representative data. Although there is no doubt that information on health consequences of a possible climate change is of interest such studies need to be based on facts rather than assumptions. Without this the manuscript does not warrant publication.
Author Response
Thank you very much for the review of our manuscript. This is an important point that we would prefer to consider but we were more interested in the perceptions of climate change as literature has indicated that it is occurring and manifests in form of events like drought, landslides, increasing malaria belt. We cited some of these in the literature but for this study, we were just doing a snapshot of what community health workers think about it.
Round 2
Reviewer 1 Report
Line 100, Spelling - recognition
Line 124, Define VHT here since it is the first use of the abbreviation in the main text of the paper. The definition is then not required on lines 146/7.
Line 170/171, Regarding my original comment - When respondents don’t know about a subject they are questioned on, I disagree with scoring this as a zero (previously state as denoting a negative response). What is your rationale for this? I recommend re-analysing the data taking this into account. I am unsure of what your mean in your reply - We did the scoring with zero so as to ease coding but we didn’t use “0” as the actual code. Scoring and coding are separate entities. At present it appears that a question such as “Do you perceive there to be more flooding due to more rains?” would score the same if the response was “I don't know” (zero) as it would if they answered “No” (zero). Any data that was scored should differentiate between these two types of response. If your comments relate to how the ‘I don’t know” responses were coded for thematic analysis this should be explained more fully to avoid confusion.
Line 184, Missing ‘the’ from sentence - in the form
Line 201, Should read rain not rainy
Line 270, Spelling - stationery
Line 302, in not into
Author Response
Thank you very much for the feedback, we have attached the comments and feedback on the review
Thanks

Reviewer 3 Report
As indicated in the authors’ response to the reviewer the survey was to elucidate capacities of village health teams in areas of Mt Elgon to investigate the efficacy of community-based health surveillance. The outcome is that there are serious deficits, and some proposals are made for improvement. If one accepts that the situation identified in few villages and few health care workers is of general interest and that an improvement is needed due to expected changes in the occurrence of diseases the manuscript may be acceptable for publication. In this case the manuscript should just focus on the specific investigation and avoid the lengthy Introduction, which presents known socioeconomic and climate dependent reasons for expected changes. The Discussion should focus on the interpretation and relevance of the data and the proposals for improvement.
Author Response
Thank you very much for the review feedback!
I have modified the methods section to indicate that the focus was on climate-sensitive health systems. This has been recommended by World Health Organization for areas where climate change plays a big role in disease surveillance. This is a pathway to integrated disease surveillance and response enhancement in Uganda. Much as this study just based on opinions/perception, we think that its a good starting ground especially for hard to reach areas like Mt Elgon.